# Cheating leads to the evolution of multipartite viruses

**Asher Leeks**[1,2,3]*, **Penny Grace Young**[2], **Paul Eugene Turner**[1,3], **Geoff Wild**[4], **Stuart Andrew West**[2]

**1** Department of Ecology and Evolutionary Biology, Yale University, New Haven, Connecticut, United States of America, **2** Department of Biology, University of Oxford, Oxford, United Kingdom, **3** Quantitative Biology Institute, Yale University, New Haven, Connecticut, United States of America, **4** Department of Mathematics, The University of Western Ontario, London, Canada

* asherleeks@gmail.com

**Data Availability Statement:** All scripts and data are available at the following Open Science Foundation repository: https://doi.org/10.17605/OSF.IO/PBE4N

**Funding:** This work was supported by: the James S McDonnell Foundation (to A.L.); the European

## Abstract

In multipartite viruses, the genome is split into multiple segments, each of which is transmitted via a separate capsid. The existence of multipartite viruses poses a problem, because replication is only possible when all segments are present within the same host. Given this clear cost, why is multipartitism so common in viruses? Most previous hypotheses try to explain how multipartitism could provide an advantage. In so doing, they require scenarios that are unrealistic and that cannot explain viruses with more than 2 multipartite segments. We show theoretically that selection for cheats, which avoid producing a shared gene product, but still benefit from gene products produced by other genomes, can drive the evolution of both multipartite and segmented viruses. We find that multipartitism can evolve via cheating under realistic conditions and does not require unreasonably high coinfection rates or any group-level benefit. Furthermore, the cheating hypothesis is consistent with empirical patterns of cheating and multipartitism across viruses. More broadly, our results show how evolutionary conflict can drive new patterns of genome organisation in viruses and elsewhere.

## Introduction

Viruses in the Nanoviridae family have genomes that are split into as many as 8 separate segments [1]. In such multipartite genomes, a successful infection only occurs if all 8 segments are present within the same host [2,3]. Multipartitism entails substantial costs because most infections will only contain a subset of the necessary genome segments and will therefore fail. Despite these costs, multipartitism has evolved independently multiple times in viruses, accounting for nearly 20% of all known viral species across at least 38 genera, including more than 40% of plant viruses (Table A in S1 Text) [4]. In other species, termed segmented viruses, the genome is similarly fragmented, but all fragments transmit inside the same viral capsid. Segmented viruses comprise approximately 9% of viral genera, including important human pathogens such as Influenza viruses [1]. Outside of viruses, other types of genome fragmentation have been found in integrated bacteriophages, plasmids, and bacterial endosymbionts

Research Council (834164 to S.A.W.); and Natural Sciences and Engineering Research Council of Canada (716206 to G.W.). The funders had no role in study design, data collection and analysis, decision to publish, or preparation of the manuscript

[1,4–8]. Why have viruses and other organisms evolved to split their genomes in this puzzling way?

Most existing hypotheses for the evolution of multipartitism rely on mechanisms that might allow viral populations to overcome the costs of being multipartite [9]. Some of these hypotheses rely on group-level benefits to multipartitism that allow a population of multipartite viruses to outcompete a population of monopartite viruses. For example, multipartitism could evolve if it allows the multipartite viral population to collectively adjust gene dosage in a host-specific manner [10,11]. Other hypotheses focus on the optimal strategy for a viral genome, assuming minimal conflict between different genome segments [12]. For example, multipartitism can evolve if smaller viral capsids survive for longer periods of time in the environment [13].

However, these existing explanations cannot explain the evolution of multipartitism in nature. It is difficult to find benefits that are large enough to overcome the substantial costs of being multipartite, especially for the 1/3 of multipartite viruses that have 3 or more genome segments [9]. For example, 1 mechanism predicts the evolution of multipartite viruses with 4 genome segments, but only if at least 100 viral particles infect each host cell [12]. This contrasts with the highest estimates from natural infections that range from 2 to 13 viral particles per host cell [14,15]. In some viral systems, mechanistic benefits, such as increased particle stability, are large enough to drive the evolution of multipartite viruses with 2 gene segments, but it is unclear whether these mechanistic advantages also exist in other groups of viruses, or other organisms that have evolved multipartitism [13]. Furthermore, evolutionary conflict is common between different viral genomes within populations; such conflict frequently destroys the kind of group optimality that these models assume, and hence we cannot use group-level advantages alone to explain the evolution of this kind of trait [16–20].

An alternative route is that multipartitism could be caused by evolutionary conflict within genomes, driven by the invasion of cheats [21,22]. In viruses, a genome is a cheat if it avoids producing a shared gene product, but can still benefit from that gene product when coinfecting cells alongside a full-length, cooperative, wild-type genome [20,23]. Cheats are common in viruses, including defective interfering genomes, which arise de novo in many kinds of viral infections [24,25]. Such cheats can arise relatively simply, via deletions in genes for replicase enzymes or capsid proteins, and can achieve extraordinarily large fitness advantages over cooperators, on the order of 10,000-times more competitive than their ancestral cooperative wild-type genomes [26]. If multiple such cheats arose, each of which lacked different gene products, then they could complement one another, replacing the wild-type virus, and resulting in the evolution of multipartitism (Fig 1) [21].

The cheat hypothesis for multipartitism may not require any population-level benefits to overcome the costs of being multipartite. Cheating could therefore explain multipartitism even if group benefits to being multipartite are weak or nonexistent and potentially even if multipartitism results in decreased productivity at the group level. Furthermore, cheating is prevalent among diverse groups of viruses, and there are multiple mechanisms by which viral cheats can gain exceptionally large fitness advantages over cooperators [20]. Therefore, the cheat hypothesis does not require any additional untested mechanistic assumptions.

However, the extent to which cheating can explain the evolution and distribution of multipartitism in viruses remains unclear. Can cheating lead to multipartitism under realistic rates of coinfection, and can it drive the evolution of multipartite viruses with more than 2 gene segments? Is cheating alone sufficient, or does it only work in combination with a group benefit to multipartitism? And can cheating still drive multipartitism even when "full cheats" emerge that encode no shared gene products at all [27]?

We examined the theoretical feasibility of the cheat hypothesis for the evolution of multipartitism, and then tested both the assumptions and the predictions of our model. We developed a game theory model that examines when mutually complementing cheats can invade and replace cooperators. This model allows us to examine the conditions that favour the initial evolution of multipartitism and to test whether multipartitism can evolve even if it reduces group-level productivity. Then, we used existing experimental data to parameterise our model, to determine whether it predicted the evolution of multipartitism under plausible conditions for different types of virus. Next, we tested the robustness of our analytical model with an agent-based simulation, which allows us to relax a number of simplifying assumptions, and examine the evolution of multipartite viruses with more than 2 genome segments. Finally, we tested the predictions of our hypothesis with an across-species comparative approach, by determining whether viral lineages that produce cheats are more likely to have evolved multipartitism.

### Analytical model

We start by considering the simplest possible case, with a full-length viral genome that encodes 2 essential gene products, both of which can be shared with other viral genomes infecting the same host cell (Fig 1) [21]. For example, one gene might encode the replicase enzyme that replicates the viral genome, while the other might encode capsid proteins that construct the viral capsid for transmission to new cells.

We allow for 3 potential strategies, each reflecting a different type of viral genome structure. The cooperator viral genome (C) encodes both genes and is the ancestral monopartite form of the virus. Two types of cheat are possible (D1 and D2), that each encode only one of the 2 genes that are present in the complete cooperator genome (Fig 1). A cell infected by both D1

## Composition of the viral population

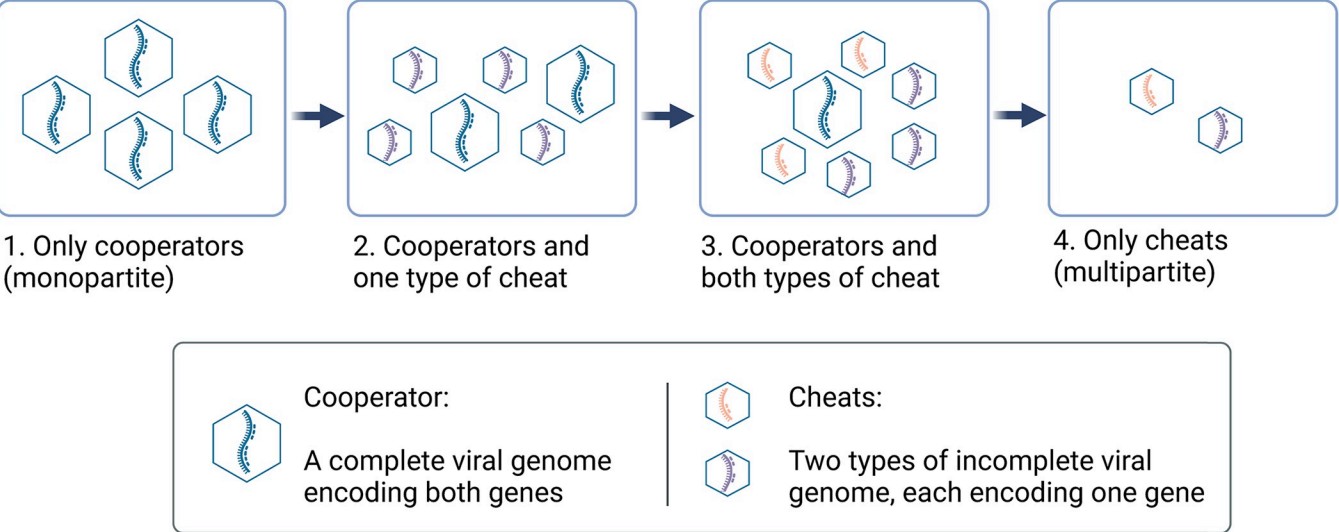

1. Only cooperators (monopartite)

2. Cooperators and one type of cheat

3. Cooperators and both types of cheat

4. Only cheats (multipartite)

Cooperator:
A complete viral genome encoding both genes

Cheats:
Two types of incomplete viral genome, each encoding one gene

**Fig 1. The evolution of multipartitism via cheating.** The ancestral monopartite population [1] consists only of cooperative viruses that each encode a full viral genome. This population is invaded first by 1 type of cheat [2], and then by a second type of cheat [3]. Each cheat has an advantage when coinfecting cells with the cooperator, and when each different type of cheat infects the same host cells, they are able to complement one another in coinfection. Consequently, provided coinfection is frequent enough, the cheats are able to drive the cooperator extinct, resulting in a multipartite population [4]. This mechanism can occur even when the final multipartite population [4] has a lower level of population productivity than the ancestral monopartite population [1]. Figure was created using BioRender.com.

and D2 would be analogous to a cell infected by the multipartite form of the virus. We assume that host cells are infected by 1 viral genome with probability 1-$\beta$ or by 2 viral genomes with probability $\beta$. In the simulation, we relax these assumptions to consider other models of coinfection, more than 2 cooperative genes, and the possibility for "full cheats" that encode no cooperative genes [27].

We assume that when a cell is infected only by a wild-type cooperator genome (C), the genome encodes both gene products, and so can successfully infect the host cell, receiving a payout $a$ (Fig 2). Biologically, this payout reflects the number of progeny viral genomes that can successfully infect a further host cell. In contrast, when a cell is infected only by a single cheat genome segment (D1 or D2), then only one of the 2 essential gene products is encoded, so the infection is unsuccessful, and the cheat genome receives a payout of 0.

If coinfection occurs, full genomes (cooperators) and partial genome segments (cheats) can interact (Fig 2). When a cooperator coinfects a cell alongside a cheat, we assume that the cheat has a replicative advantage, and therefore, the cooperator receives a smaller payout $c$, while the cheat receives a larger payout $b$. If 2 cheats of the same type coinfect the same cell, then only one of the 2 gene products is produced, and so the infection aborts and both receive a payout

## (a) Social interactions in infected cells

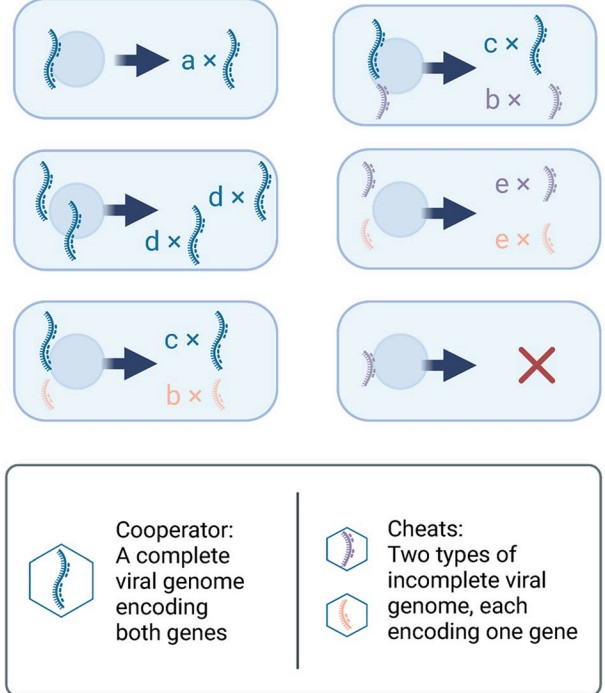

## (b) Fitness consequences of social interactions (payoff matrix)

|  | C | D1 | D2 |
|---|---|---|---|
| Cooperator (C) | d | c | c |
| Cheat 1 (D1) | b | 0 | e |
| Cheat 2 (D2) | b | e | 0 |

## (c) Equations describing the model

$$W(C) = a\,(1\text{-}\beta) + \beta\,(c\,q + c\,z + d\,p)$$

$$W(D1) = \beta\,(b\,p + e\,z)$$

$$W(D2) = \beta\,(b\,p + e\,q)$$

**Fig 2. The costs and benefits of cheating.** (a) We assume that the expected number of viral genomes produced in an infected cell depends on whether cells are infected by cooperators (that encode both genes), cheats (that each encode one gene but not the other), or both. (b) The number of progeny viral genomes of each type can be captured in a payoff matrix. Here, each entry reflects the number of viral progeny that each strategy on the left (rows) receives when it coinfects a cell alongside the strategy listed at the top (columns). (c) We analyse the dynamics of the payoff matrix using replicator dynamics, yielding simple equations for the change in relative frequency of cheats and cooperators (Eq 1 in main text). Figure was created using BioRender.com.

0. However, if one cheat of each type infects the same cell, then there can potentially be complementation leading to the production of viral progeny, and so they each receive a payout $e$. Cooperators receive a payout $d$ when coinfecting cells alongside another cooperator, in order to allow for the possibility that cooperators do better (or worse) when they coinfect cells alongside other cooperators [28–30].

Our model allows us to consider a wide range of biological scenarios. The values summarised in the payoff matrix of Fig 2 represent the number of successful progeny viral genomes that can infect a further host cell. Consequently, these values could reflect a number of different biological mechanisms, such as increased burst size, faster replication speed, or increased particle longevity [11–13]. It is possible for multipartitism to provide a group-level benefit in this model if cells infected by both types of cheat are more productive than cells infected by 2 cooperators ($e > d$).

We denote the relative frequency of cooperator viral genomes (C) as $p$, of cheats of the first type (D1) as $q$, and cheats of the second type (D2) as $z$ ($p + q + z = 1$). The fitness ($W$) for each type of viral genome in a large, well-mixed population of susceptible host cells is:

$$W(C) = a\,(1 - \beta\,) + \beta\,\,(c\,q + c\,z + d\,p) \qquad 1.1$$

$$W(D1) = \beta\,\,(b\,p + e\,z) \qquad 1.2$$

$$W(D2) = \beta\,\,(b\,p + e\,q). \qquad 1.3$$

In the case where all cells are doubly infected ($\beta =1$), and with just 1 type of cheat (only D1 or D2), this model becomes the Hawk-Dove or Snowdrift game (since $b > d > c > 0$) [31–33].

**Multipartitism can evolve under realistic conditions.** We consider a transition to multipartitism to have occurred if the 2 types of cheat are able to invade and then completely replace the monopartite cooperator. We found that each type of cheat can invade provided the benefits to cheating ($b$) are sufficiently large, and coinfection ($\beta$) is sufficiently common ($\beta > \frac{a}{a+b-d}$) (Methods; Figs A and B in S1 Text). We then find that cheats can fully replace the cooperators provided the possibility for complementation between cheats ($e$) is larger than a threshold minimum value: $e^* = \frac{2a-2a\beta+2c\beta}{\beta}$.

Multipartitism evolves more easily when:

1. The probability of coinfection ($\beta$) is higher, because a cheat is more likely to be in a cell that contains either a cooperator (that it can exploit) or a complementary cheat.

2. The advantage from cheating ($b$) is higher, making it easier for cheats to invade.

3. Cheats are better able to complement each other (higher $e$), meaning that mixtures of the 2 types of cheat are better able to replicate, and therefore, replace cooperators.

4. Cooperators are strongly outcompeted by cheats within the same cell (low $c$).

5. Cooperators gain a larger benefit from coinfecting cells alongside other cooperators, compared to infecting cells alone ($d > a$).

Our model predicts that multipartism is favoured more easily than previous hypotheses. In particular, we find that multipartitism can evolve at much lower coinfection rates, especially when the potential for group-level advantages ($e$) is low. For example, our condition for the evolution of multipartitism ($e^* \geq \frac{2a-2a\beta+2c\beta}{\beta}$) predicts that multipartitism can evolve when as few as half of cells are coinfected ($\beta =0.5$), even in the absence of any group benefits to being multipartite ($e = d$), provided that cheats are highly competitive relative to cooperators ($c\sim =$

$0$), and cooperators gain large benefits from interaction with other cooperators ($a \leq \frac{d}{2}$) (Methods) [28–30]. How commonly are permissible conditions such as these found in real viruses?

The existing empirical data support our hypothesis that cheating can lead to the evolution of multipartitism relatively easily (Fig 3 and Table B in S1 Text). We used published data to estimate the parameters in our model for 6 species (Methods; Table B in S1 Text). Studies examining cheating by defective interfering genomes typically observe that cooperators perform badly when coinfecting with cheats ($c\sim = 0$) (Table B in S1 Text). For such cheats, our model predicts that multipartitism can evolve without any group benefit to being multipartite ($e = 1$), and when as few as 50% of cells were coinfected ($\beta = 0.5$), which occurs when an average of 1.7 viral genomes infect each host cell (MOI = 1.7). However, the extent to which cheating favours multipartitism depends on the type of cheat. The cheating benefit provided by point mutations was much smaller, such that this form of cheating would be unlikely to favour the evolution of multipartitism (Fig 3 and Table B in S1 Text).

**Multipartitism does not require group benefits.**   We have hypothesised that the evolution of multipartitism can be favoured by cheating, even if it leads to a lower population-level productivity. We can investigate this possibility theoretically, by comparing the productivity of the original monopartite population with the productivity of the subsequent multipartite population. The multipartite population has a lower productivity than the ancestral monopartite population ($W(\mathrm{C})_{p \to 1} > W(\mathrm{D1})_{z \to 1/2, p \to 0, q \to 1/2}$) provided the possibility for complementation is low enough ($e < e{**} = 2\left(d + a\left(-1 + \frac{1}{\beta}\right)\right)$) (Methods). Because

$e{**} > e{*}\left(2\left(d + a\left(-1 + \frac{1}{\beta}\right)\right) > \frac{2a - 2a\beta + 2c\beta}{\beta}\right)$, it is always possible for cheat complementation (e) to be large enough that the transition to multipartitism occurs (e > e*), but also small enough that the resulting multipartite population has a lower productivity than the original monopartite population (e* < e < e**). This finding is consistent with an existing model of multipartitism, which can be captured as a special case of our model [21]. Our model extends this finding and demonstrates that group benefits are never required for multipartitism to evolve.

## Simulation

To extend our analytical findings and to check their robustness, we also wrote an agent-based simulation in which we relaxed some simplifying assumptions (Methods). We assumed a finite number of host cells, which could each be infected by a variable number of viral particles drawn from a Poisson distribution, to reflect different multiplicities of infection (MOI). We allowed the viral genome to contain up to 8 genes, and cheat strategies that lacked any number of these genes, including "full cheats" that encoded zero genes whatsoever [27]. We later extended our simulation to consider the possibility that multiple viral genomes can be encapsidated inside the same viral particle.

When considering the evolution of multipartite viruses with 2 genome segments, our simulation found broadly similar results to the analytical model, with coinfection and complementation favouring the evolution of multipartitism, and group benefits not being required (Fig C in S1 Text). Previous studies have suggested that the presence of "full cheats," which encode no genes whatsoever, can prevent multipartitism evolving by cheating [27]. In contrast to this suggestion, we found that such full cheats could slightly reduce the likelihood that multipartitism evolved, but that multipartitism still evolved relatively easily.

Our simulation also predicted that different combinations of partial cheat types could coexist, reflecting the evolution of multipartite viruses with different numbers of genome segments (Fig 4). As we allowed higher numbers of genes, the number of possible cheating strategies increased considerably—for instance, with 3 genes, there are 8 possible strategies (if genes are

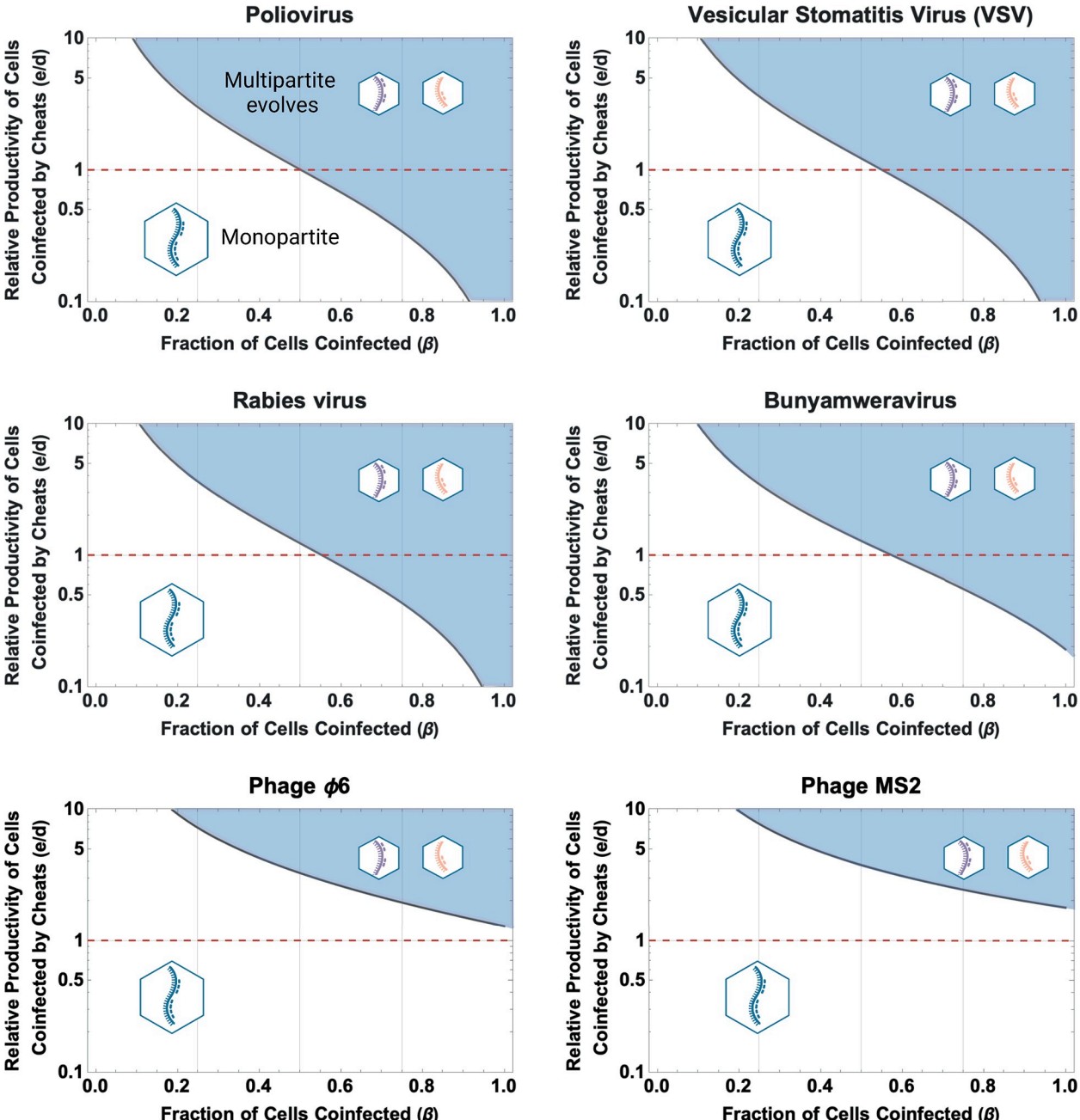

**Fig 3. Cheats can drive the evolution of multipartitism under realistic conditions.** We used existing experimental data to derive estimates for the parameters in our analytical model (Methods; Table B in S1 Text). We then used these parameters to determine whether our model would predict the evolution of multipartitism. We plot the fraction of cells infected by multiple viral genomes ($\beta$) on the x-axis, and the minimum productivity of cells coinfected by cheats, relative to cells infected by cooperators ($e/d$) on the y-axis. In the shaded regions, our model predicts that multipartitism evolves; in the unshaded regions, the population remains monopartite. The top 4 panels provide examples of species with defective interfering genomes: poliovirus, vesicular stomatitis virus, rabies virus, and Bunyamweravirus. In such species, cheating can favour the evolution of multipartitism when as few as half of all cells are coinfected, even when there is no benefit to being multipartite ($e/d \leq 1$; highlighted by the dashed red line). The bottom 2 panels provide examples of cheats derived from point mutations or small deletions: Phage Φ6 and Phage MS. In such species, our model predicts that the evolution of multipartitism requires both higher rates of coinfection to evolve and some group benefit to multipartitism. This figure can be generated using the data and code at https://doi.org/10.17605/OSF.IO/PBE4N.

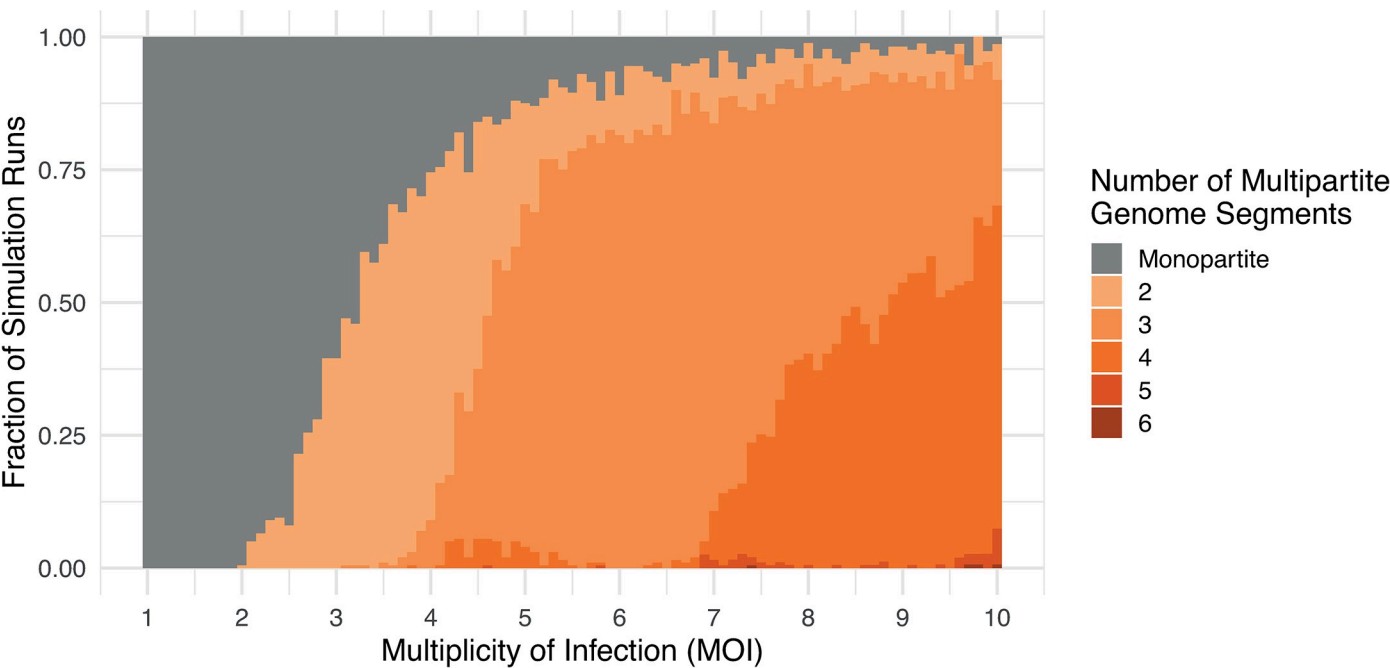

**Fig 4. Cheating drives the evolution of multipartite viruses with more than 2 genome segments.** This figure plots the cumulative fraction of simulations that led to different numbers of multipartite genome segments, for a viral genome containing 8 genes. Splits to higher numbers of genome segments (indicated by darker orange) were more likely when a larger average number of viruses infected each host cell (a higher multiplicity of infection or MOI; λ in our model). Each vertical bar represents 500 simulation runs over 10,000 generations. This figure can be generated using the data and code at https://doi.org/10.17605/OSF.IO/PBE4N.

each present (1) or absent (0), the potential strategies are: 111, 110, 011, 101, 001, 010, 100, and 000). We found that when there were more genes, and so more possible cheating strategies, the simulation could stochastically reach different equilibria at which different combinations of partial cheat types coexisted (Fig 4).

**Cheating can produce multipartite viruses with more than 2 segments.** Our simulation showed that cheating could drive the evolution of multipartite viruses with more than 2 genome segments, even when coinfection rates were relatively low (Fig 4). For example, consider the case where the viral genome contained 8 genes, with no group benefits to multipartitism, and when cells were infected with an average of 5 viral particles. In this case, we found that multipartite viruses with 3 or 4 genome segments evolved in nearly 3 quarters of the simulation runs (Fig 4). At such relatively low levels of coinfection, these tri- and quadri-partite viruses had drastically lower levels of population productivity than the ancestral monopartite populations, because a large fraction of cells were only infected by a subset of the required genes, which leads to the infection failing.

Consequently, the cheat hypothesis for multipartitism requires coinfection rates that fall well within the range that are found in nature and that are orders of magnitude lower than those required by existing hypotheses [12,14] (Fig 3). While the cheat hypothesis does not require group benefits, it could potentially interact with group benefits. We tested this and found that group benefits increased the likelihood of multipartitism evolving at low rates of coinfection and when there were few genes in the genome (Fig D in S1 Text). However, we found that even large group benefits ($e = 2$) made little difference to the pattern of multipartitism at higher rates of coinfection or with larger genomes, further emphasising that cheating can drive multipartitism without group benefits.

We found that multipartitism was more likely to evolve when the viral genome contained a greater number of genes encoding shared gene products (social genes) (Fig E in S1 Text). We assumed that the fitness of a cheat depends on the fraction of the total number of genes it encodes. Consequently, when there are more genes in the genome, full cheats, which encode no genes, have a smaller fitness relative to partial cheats, that still encode one or more cooperative genes. This means that as the number of genes in the viral genome increases, it becomes more likely that the simulation reaches a multipartite equilibrium, with only partial cheats, and less likely that the simulation reaches a monopartite equilibrium, with cooperators and full cheats. These results make multipartitism easier to explain: multipartitism evolves more easily when multiple types of cheat are possible, each cheating a different shared gene product.

We also used the simulation to explore how additional details of viral genome architecture would influence the evolution of multipartitism. We found that multipartitism was more likely to evolve when cheats that lacked larger numbers of genes gained proportionately smaller competitive advantages (decelerating advantages to a shorter length) (Fig C in S1 Text).

**Multipartite viral populations are more resistant to invasion by full cheats.**   We have focused our model on the question of when cheating drives the evolution of multipartite genomes without any group benefits to multipartitism. However, in our simulation, we found that a new kind of group benefit to being multipartite emerged that has not previously been suggested. When the viral population became multipartite, the abundance of "full cheats," which encode no genes at all, decreased. The abundance of full cheats decreased further when the viral population split into greater numbers of fragments—for instance, tripartite viruses experienced fewer full cheats than bipartite viruses (Fig 5). The details of which genes were encoded on which genome segments also mattered: multipartite viruses with more uneven splits (such as 1 genome encoding 7 genes and another encoding 1) were less exploited than multipartite viruses with more even splits (such as both genomes encoding 4 genes each) (Fig 5).

This benefit emerged because we assumed that genome strategies were more competitive within the cell if they encoded fewer genes. In a monopartite population, full cheats compete against full-length viral genomes for cellular resources, and hence gain a large share. However, in a multipartite population, full cheats compete against smaller genome fragments for cellular resources, and hence gain a smaller share. The greater the number of genome fragments, the fewer genes each fragment encodes, and so the more competitive each fragment becomes compared to a full cheat. A similar logic explains why the genome evenness matters: when one genome fragment is smaller than the other, it reaches a higher equilibrium frequency within the population, meaning that cells tend to be infected by many copies of the smaller fragment, making them a more competitive environment that is harder for full cheats to exploit.

**Cheating can favour the evolution of segmented viruses.**   Our analytical model assumed that each viral genome is encapsidated inside a separate viral particle, reflecting the biology of multipartite viruses. However, in segmented viruses, each genome fragment is encapsidated inside the same viral particle [34]. Can cheating also drive the evolution of segmented viruses, when each genome segment is encapsidated inside the same virion?

To answer this question, we extended our simulation to allow for the possibility that multiple viral genomes are packaged inside the same virion. We introduce a new parameter, "prop_single," which controls the likelihood that each virion contains just 1 viral genome; "prop_single" fraction of viral particles contained 1 viral genome and "1-prop_single" viral particles contained 2 viral genomes.

We found that when multiple viral genomes were packaged inside the same virion, the population evolved genome fragmentation more easily (Fig 6). This occurred because when virions could contain multiple genomes, infected cells infected by a single virion could contain

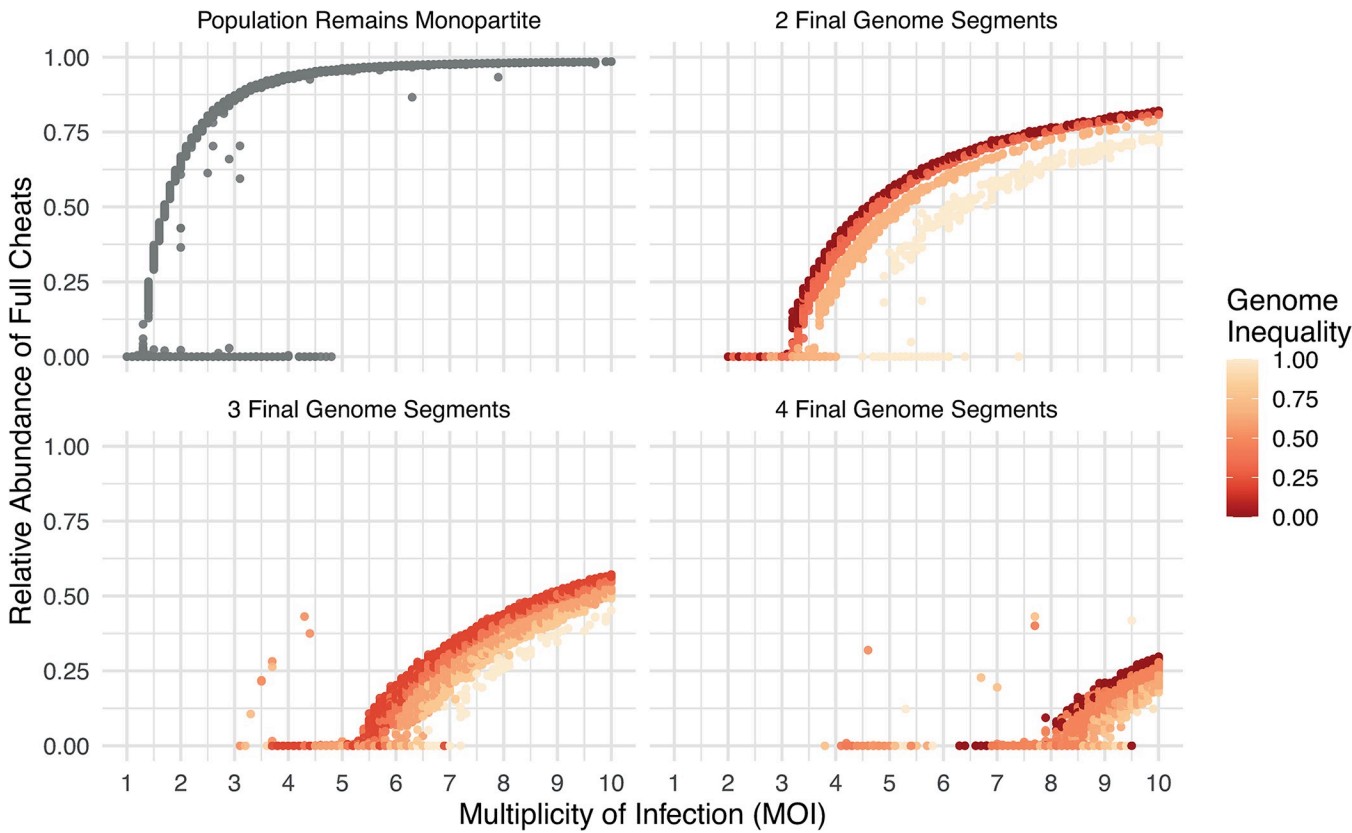

**Fig 5. Multipartite viruses were more resistant to exploitation by full cheats.** Here, we plot the relative abundance of "full cheats" that encode no genes whatsoever, across simulation runs that resulted in multipartite viral genomes with different numbers of genome segments. We found that when multipartitism evolved, the viral population was subsequently less exploited by full cheats. This effect was stronger for multipartite viral populations with higher numbers of genome segments and for multipartite viruses with more uneven distribution of genes across their genome segments (lighter orange shading). Each point represents an individual simulation run for a viral genome containing 8 genes. This figure can be generated using the data and code at https://doi.org/10.17605/OSF.IO/PBE4N.

multiple genomes. Hence, a world in which virions contain multiple genomes means that the multiplicity of infection in terms of the number of genomes is higher than the multiplicity of infection at the level of number of viral particles. Hence, genome fragmentation evolved when fewer viral particles infected each host cell (a lower multiplicity of infection or MOI) (Fig 6).

## Comparative predictions

Finally, we collected data across viruses, to test our hypothesis that cheating favours the evolution of multipartite viruses. This hypothesis predicts that the evolution of multipartitism is more likely in viruses where cheating occurs, and especially when cheats substantially outcompete cooperators within cells, as in the case of defective interfering genomes (Methods).

Our ability to test this prediction is limited by several factors, especially a lack of consistent sampling for defective interfering genomes, and the fact that viruses do not share a single phylogenetic tree [35,36]. Consequently, we carried out a relatively conservative analysis, comparing across virus Realms. Each virus Realm is thought to represent an independent evolutionary origin of viruses. Across these realms, we found that the fraction of genera known to contain multipartite viruses was positively correlated with the fraction of genera known to produce defective interfering genomes (Fig 7). These results are consistent with the hypothesis that cheating has driven the evolution of multipartitism across viruses.

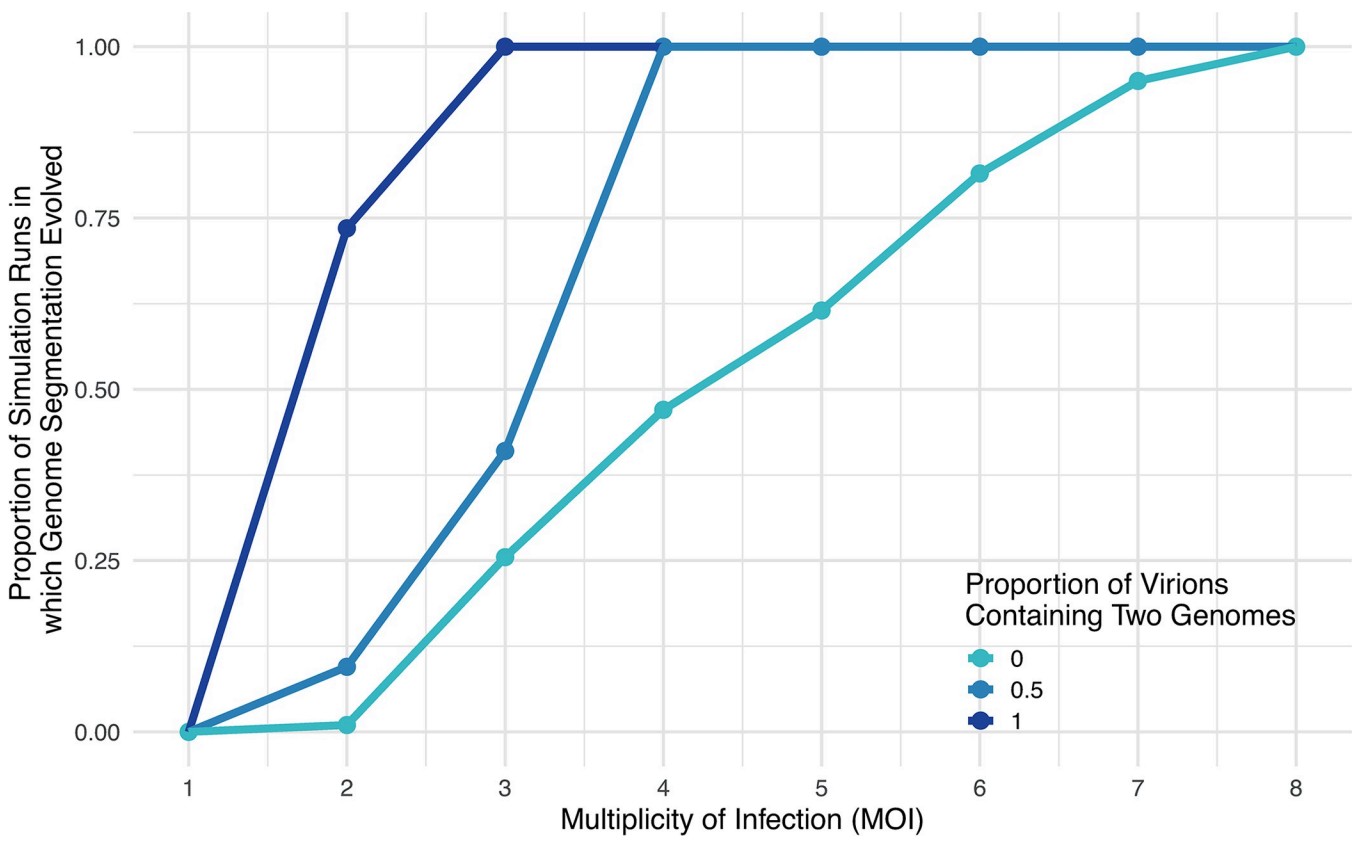

**Fig 6. Cheating can favour the evolution of segmented viral genomes.** We extended the simulation to allow multiple viral genomes to be packaged inside the same virion. Genome fragmentation evolved at lower multiplicities of infection when multiple genomes were packaged inside the same virion (reflecting the evolution of segmented viruses), than when genomes were packaged inside separate virions (reflecting the evolution of multipartite viruses). Each point represents the fraction of 100 simulation runs in which the full-length cooperator was driven extinct and replaced with complementing sets of viral cheats (genome fragmentation). This figure can be generated using the data and code at https://doi.org/10.17605/OSF.IO/PBE4N.

## Discussion

We found that cheating can lead to the evolution of multipartitism under realistic conditions (Figs 1 and 3). When cheats emerge that lack key genes, but that can complement one another in coinfection, these cheats can outcompete and then replace cooperators, resulting in a multipartite population in which the complete genome is distributed across multiple separately encapsidated partial genomes (Fig 2). Cheating can drive the evolution of multipartite viruses with more than 2 genome segments under levels of coinfection that are readily found in natural viral infections (Fig 4), and without any group benefits (Fig 3). By parameterising our models with existing experimental data from different viruses, we find that multipartitism can evolve far more easily via cheating than via other suggested mechanisms (Fig 3). Cheating can also favour the evolution of segmented viruses, where different segments are encapsulated together (Fig 6). In support for our theory, we found that virus realms in which cheating is more common have higher rates of multipartitism (Fig 7). Overall, and in contrast to existing theory, our results suggest that multipartitism does not need to be a group-level adaptation and can instead arise as the endpoint of evolutionary conflict between genomes.

### Cheating drives the evolution of multipartitism in viruses

We found that multipartitism could evolve under a far broader range of conditions than predicted by other models of multipartitism in viruses, and with fewer specific assumptions about

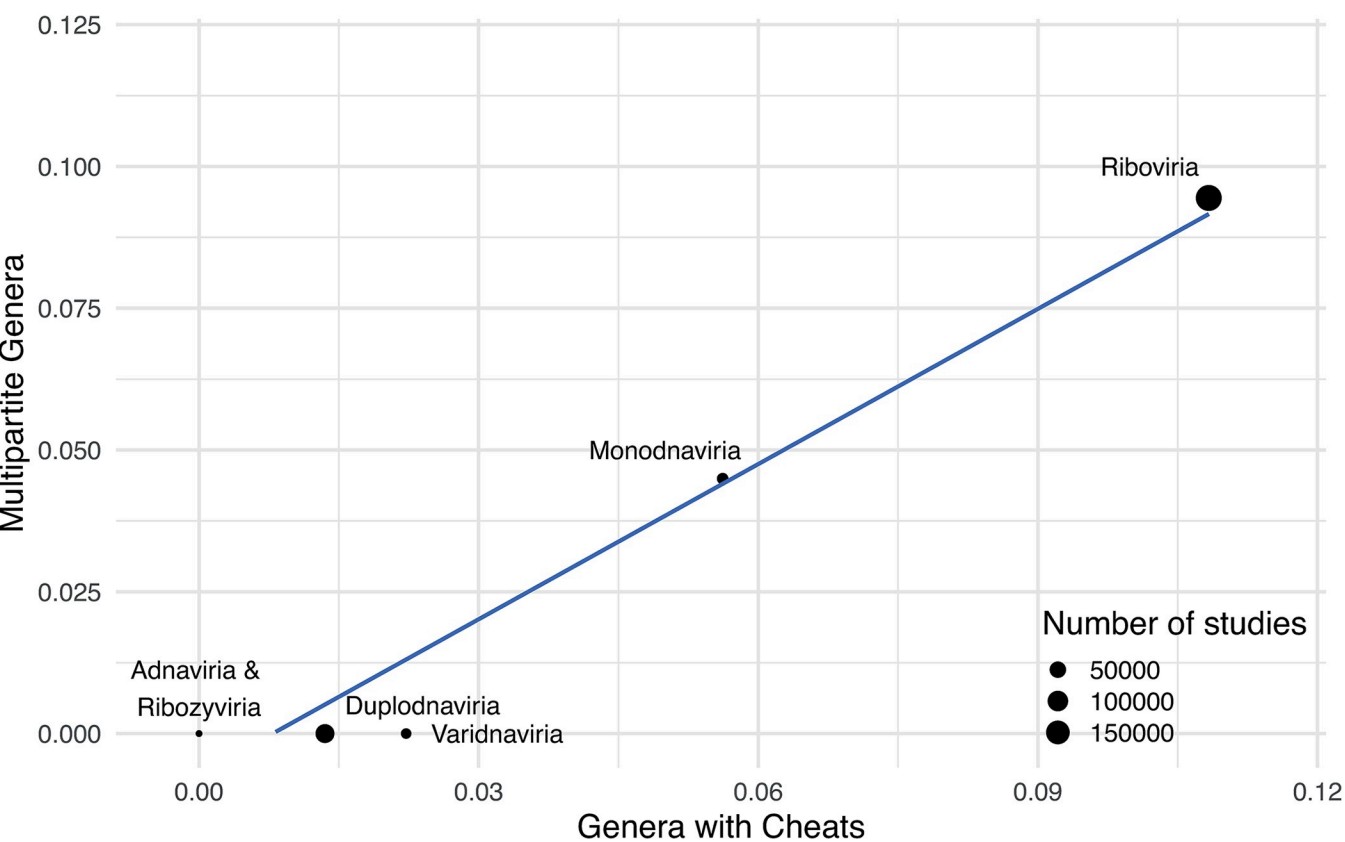

**Fig 7. Cheating is associated with multipartitism across the virosphere.** The fraction of genera known to contain defective interfering genomes is plotted against the fraction of genera known to be multipartite for 4 viral Realms, where each Realm represents a likely independent origin of viruses (Methods). This figure can be generated using the data and code at https://doi.org/10.17605/OSF.IO/PBE4N.

viral biology (Figs 3 and 4). While our model is consistent with previous work in showing that group benefits can help multipartitism, our model does not rely on a group benefit [9,11,12,37]. Hence, our results agree qualitatively with other models when group benefits are very high, but diverge by orders of magnitude when group benefits are low (Fig 3). Our results also differ from other models for the evolution of multipartite viruses with more than 2 genome segments, which we find can evolve under far lower rates of coinfection. For example, we find that multipartite viruses with 3 or 4 genome segments frequently evolve when as few as 4 or 5 viral particles infect each host cell (Fig 4), falling well within empirical estimates of coinfection within plants [14,15]. In contrast, existing models developed to explain multipartitism tend to require coinfection rates that are at least 1 or 2 orders of magnitude higher (Fig 4) [9,12]. Cheating may therefore be particularly important for the 1/3 of multipartite genera that have 3 or more gene segments (Methods; Table A in S1 Text).

It is easier for multipartitism to evolve in our model because cheating can lead to multipartitism even when this is costly for the viral population, provided that cheats gain sufficiently large advantages over cooperators [21]. We have shown that a sufficient advantage to cheating exists for a range of viruses (Fig 3), but we also expect this to hold more broadly, because there are many well-described mechanisms that can give cheats particularly large advantages over cooperators [20]. For example, viral cheats are often shorter than cooperators, resulting in disproportionate advantages in replication speed, especially when viral genome replication is geometric or involves nonlinear feedbacks [30,38,39]. Beyond being shorter, viral cheats can also

**Table 1. Some testable predictions arising from the model.**

| Prediction | Logic | New data required to test |
|---|---|---|
| Viral lineages that produce defective interfering genomes are more likely to evolve multipartitism | Cheating drives multipartitism under permissive conditions (Figs 3 and 4), supported by current data (Fig 7) | - A well-resolved viral phylogeny<br>- Data on cheating and multipartitism for a range of viruses sufficient to conduct ancestral state reconstructions |
| Viruses with more genes in their genomes should evolve multipartitism more easily | When there are larger viral genomes, full cheats gain a smaller advantage relative to partial cheats (Fig E in S1 Text) | - Experimental evolution of multipartitism in viruses with different genome sizes<br>- A comparative test on whether viral lineages with larger genomes are more likely to evolve multipartitism |
| Viruses with genes of different lengths should evolve multipartitism more easily | When viral genes are different lengths, full cheats gain a smaller advantage relative to partial cheats (Fig C in S1 Text) | - Experimental evolution of multipartitism in viruses with different genome compositions<br>- A comparative test on whether viral lineages with more heterogeneous genomes are more likely to evolve multipartitism |
| Multipartite viruses should have a lower population fitness than ancestral monopartite viruses | Cheating can drive multipartitism, resulting in a less productive viral population | - Multipartite viral lineages could be transient on an evolutionary scale, characterised by multiple short phylogenetic branches |
| Multipartite viruses should be less vulnerable to cheating | Full cheats are less likely to invade multipartite viral populations (Fig 5) | - A comparative study testing whether multipartite viral lineages are less affected by cheats than sister lineages that remained monopartite |

gain advantages by modifying their genomes in ways that cooperative viruses cannot, such as in cheats of Poliovirus and phage MS2, which gain disproportionate entry into virions, or in cheats of filamentous phage f1, which insert additional sequences to attract replicase enzymes [26,40,41]. Consequently, our work does not require new kinds of mechanism to be uncovered in order to explain multipartitism, nor does it require searching for elusive group benefits to multipartitism. Instead, we suggest that a fruitful direction for empirical work is to identify conditions that allow for moderately high levels of coinfection, combined with multiple genes that can each be cheated.

## New empirical predictions

Our analyses make several specific testable predictions about the evolution of multipartite viruses (Table 1). Firstly, our central comparative prediction, which viral lineages with higher levels of cheating will be more likely to evolve multipartitism, could be tested more formally (Fig 7). This would require more viral phylogenies, and more data on which viruses are multi-partite and/or produce cheats. Secondly, we predict that multipartitism is more likely to evolve when the genome contains many genes that can be complemented *in trans* (social genes), and especially if these genes are different lengths, since in that case there are more likely to be decelerating advantages to lacking additional genes (Fig C in S1 Text). Thirdly, in contrast with most existing models of multipartitism, our findings show that group benefits are not required, and so multipartite viruses could have a far lower population-level productivity than their monopartite ancestors. If this is the case, we might expect multipartite viruses to be relatively transient on an evolutionary timescale, characterised by multiple short phylogenetic branches that represent extinct lineages, analogous to asexual lineages in eukaryotes [42].

## Group benefits and the evolutionary maintenance of multipartitism

Our model suggests that group-level benefits are not needed for the origins of multipartitism, but that they could play a role in the maintenance of multipartitism over evolutionary time-scales. One possibility is that there are intrinsic benefits to being multipartite in certain systems. For example, in foot-and-mouth disease virus, smaller virions survive for longer in the environment and this provides an intrinsic benefit to the multipartite viral population [13].

Alternatively, multipartitism may allow for the simultaneous transcription and translation of viral gene products, allowing multipartite viruses to use cellular resources more effectively. In our models, we have captured these kinds of "intrinsic" benefits to multipartitism in a relatively simple way by assuming that host cells infected by multipartite viruses produce a greater number of infectious virions, determined by our parameter $e$ (Methods). However, future work could consider more complex ways in which these advantages manifest. For example, the extent to which our "e" parameter captures the benefit from simultaneous transcription and translation may depend on details that will vary between different viral systems, such as the extent to which cellular resources are limiting, the relative benefits of more abundant viral gene products, and whether these advantages can be gained through other mechanisms, such as the synthesis of subgenomic mRNAs.

Alternatively, the evolution of multipartitism could open the door to further adaptations that may not have been possible for the monopartite ancestor. Many existing hypotheses for multipartitism fall into this category, such as models that assume that multipartite viruses can dynamically adjust their gene dosage to adapt to new host types, and experimental studies that show an advantage to changes in the ratios of genome segments [11,43]. We found that a similar kind of evolved benefit emerged in our models: when multipartitism evolves, full cheats, which encode no genes at all, are less able to exploit the resulting viral population. This reduction in cheat load could represent a new kind of group benefit to multipartitism, analogous to how pleiotropy can offer benefits to bacteria by constraining the emergence of cheats [44]. Future work could consider further scenarios in which this group benefit may be important, such as in competition with viral satellites or with other species of virus. More broadly, multipartitism could be analogous to other complex social traits, such as eusociality, sexual reproduction, and reproductive division of labour, where it is useful to distinguish between the origin and maintenance [42,45–47].

## The natural history of multipartite and segmented genomes

We have shown that cheating can drive the evolution of genome fragmentation, leading to both multipartite viruses, in which each genome fragment transmits independently, and segmented viruses, in which genome fragments transmit together. Our model is consistent with empirical patterns of cheating across viruses, but to what extent can we explain the natural history of segmented viruses, multipartite viruses, and related genome fragmentation phenomena?

Most known multipartite viruses infect plants (Table A in S1 Text). This observation could be consistent with cheating driving multipartitism, since many features of plant viral infections match the conditions under which our model predicts the evolution of multipartitism. For example, plant viral infections are often long lasting, chronic, and involve vertical transmission from parents to offspring; these factors could allow sufficient viral generations for cheats to replace cooperators, even if this replacement is costly for the viral population as a whole (Fig A in S1 Text and Fig 3) [48,49]. Similarly, plant viruses often transmit via insect vectors, which can sometimes transmit relatively large numbers of virions to new hosts, especially when multiple individual insect vectors sequentially transmit virions to the same host plant [3,50]. These observations are further consistent with the fact that in most viruses, cheats appear to primarily spread within but not between hosts, whereas in plant viruses, widespread satellite virus cheats consistently transmit between hosts [20,51]. Consequently, if cheating does drive the evolution of multipartite viruses, this could explain why multipartitism appears to be so common in plant viruses.

In contrast, segmented viruses, in which all genome fragments transmit inside the same virion, are common across a wide range of viruses, not just plant viruses. This is consistent

with our model predictions, which find that segmented viruses can evolve at a lower multiplicity of infection than multipartite viruses (Fig 6). Furthermore, segmented viruses may evolve more easily when narrow between-host bottlenecks occur, since coinfection can occur even when a small number of virions are transmitted. Our results suggest a number of further questions. Under what conditions are multipartite viruses favoured over segmented viruses? Are viral cheats and cooperators under different selection pressures to undergo collective transmission [52]? To what extent does genome segmentation depend on details of the viral capsid, such as whether it can expand to contain additional genomes?

Genome fragmentation also occurs outside the viral world. *Hodgkinia* symbionts of cicadas, some plasmids, and a number of temperate phages, have evolved genomes that are also fragmented, depending on multiple mutually complementing sets [5–8]. Some of these examples, such as *Hodgkinia* symbionts, may be more analogous to segmented viruses, since all genome fragments transmit together vertically. Other examples may be more analogous to multipartite viruses, depending on their lifecycle. For example, temperate phages transmit together when reproducing vertically via bacterial cell division (analogous to segmented viruses), but transmit separately when reproducing horizontally via horizontal gene transfer (analogous to multipartite viruses). In all of these cases, there is the potential for social interactions between fragments. We have shown that when such social interactions are possible, cheating can drive the evolution of both multipartite genomes (in which each genome fragment transmits separately) (Fig 3) and segmented genomes (in which all genome fragments transmit together) (Fig 6). Therefore, cheating could be the driving force behind genome fragmentation across the tree of life. Genome fragmentation may be particularly common in viruses because viral cheats appear to be more common, and gain larger fitness advantages, than cheats in other systems [20].

## Conclusion

Previous work has shown that evolutionary conflict can lead to manipulation at all levels of biology, from genomes to societies [18,46,53]. This has resulted in a diverse and complex range of systems designed to control such conflict, such as fair meiosis to prevent the spread of selfish genes, uniparental inheritance of symbionts, or policing mechanisms to suppress cheats [54–58]. We have shown that conflict can also act as a destructive force, fragmenting the genome, in a stark example of selfish genetic interests trumping those of the group.

## Methods

### Analytical model analysis

In order for multipartitism to evolve via cheating, a population initially composed of monopartite cooperative viruses must be invaded by both types of cheat, and then, the cheats must drive the monopartite cooperators to extinction. To find conditions under which these steps take place, we carry out evolutionary stability analyses on our fitness Eqs 1.1–1.3. For these analyses, we assume that p+q+z = 1, $0 \leq \beta \leq 1, d \geq a > c > 0, e \geq 0, b > 0$.

Firstly, each cheat can invade a population initially consisting of cooperators when it has a higher fitness when rare than the resident cooperators. This is given by $W(\text{D}1)_{p \to 1, q \to 0, z \to 1-p-q} > W(\text{C})_{p \to 1, q \to 0, z \to 1-p-q}$, which is satisfied when $b > \frac{a(1-\beta)+d\beta}{\beta}$; verbally, a cheat can invade provided coinfection is sufficiently common ($\beta$ is relatively high) and provided it gets a relatively large benefit when coinfecting with a cooperator ($b$) relative to the benefit a cooperator gets when infecting cells ($d$ in coinfection and $a$ in single infection). Once a cheat has invaded, we can find the equilibrium frequency that the cooperator and cheat reach by finding when $W(\text{D}1)_{z \to 1-q-p, q \to 1-p} = W(\text{C})_{z \to 1-q-p, q \to 1-p}$. This is satisfied provided our

previous conditions for a cheat to invade are met ($b>d$ and $\beta > \frac{a}{a+b-d}$), at which point the equilibrium frequencies of the cooperator and cheat respectively are $p^* = \frac{a-a\beta+c\beta}{\beta(b+c-d)}$ and $q^* = 1-p^*$. At this equilibrium, the relative frequency of the cheat increases as coinfection ($\beta$) becomes more common, and as the relative benefit to cheating ($b$) increases, relative to the payoffs that cooperators get when infecting cells ($d$ and $a$).

A second cheat can then invade this equilibrium provided it has a higher fitness than the resident cheats or cooperators do when it is rare; this is given by $W(\text{D2})_{z\to0,p\to p^*,q\to1-p^*} > W(\text{D1})_{z\to0,p\to p^*,q\to1-p^*}$. This occurs given the additional requirement that $e > 0$; that is, the second cheat can invade provided at least some complementation is possible between the 2 types of cheat genome. Once the second cheat invades, the following equilibria are reached: $p^* = 1-2\,q^*$ and $q^* = z^* = \frac{a-(a+b-d)\beta}{(-2(b+c-d)+e)\beta}$. At this equilibrium, the 2 types of cheat have the same frequency, because they have symmetrical payoffs (Fig 2); the frequency of the cooperators decreases as coinfection ($\beta$) becomes more common, and as the relative benefit of cheating (b) or complementation between cheats (e) increase.

Next, we can find when the population of cooperators is fully replaced by the complementing cheats by finding when $p^* = 1-2\,q^* \to 0$, which occurs when $e \geq \frac{2a-2a\beta+2c\beta}{\beta}$. That is, the cheats are able to fully replace the cooperators provided the possibility for complementation between cheats (e) and the rate of coinfection ($\beta$) are high enough.

Complementation between cheats (e) is important for the evolution of multipartitism because it reduces the strength of negative frequency dependence (Fig A in S1 Text). When there is no complementation (e = 0), our model broadly follows the dynamics of a Hawk-Dove game, in which cheats (Hawks) cannot drive cooperators (Doves) extinct, because when cooperators are very rare, cheats mostly interact with other cheats, and receive a payout of zero [31–33]. However, the inclusion of the second type of cheat in our model changes these dynamics, because now, when cooperators are rare, cheats do not necessarily meet themselves, but can instead meet the other type of cheat and receive a fitness payout e. Consequently, when e is large enough, the negative frequency dependence that usually maintains cooperators in the population is overcome and the cheats completely replace the cooperators (Fig A in S1 Text). Our e* condition gives the minimum degree of complementation that would allow cheats to replace cooperators in this way.

If complementation between cheats is high enough for cheats to replace cooperators, does this imply the resulting cheat population is fitter than the ancestral cooperator-only population? We can test whether this transition to multipartitism results in such a group-level fitness increase by comparing the mean fitness of the cooperators at the equilibrium containing no cheats, with the mean fitness of the cheats at the equilibrium containing no cooperators. The multipartite equilibrium has a lower mean fitness than the monopartite equilibrium provided $W(\text{C})_{p\to1} > W(\text{D1})_{z\to1/2,p\to0,q\to1/2}$, i.e., $a(1-\beta) + d\beta > \frac{e\beta}{2}$, which is satisfied when $e < 2\left(d + a\left(-1 + \frac{1}{\beta}\right)\right)$. This new e condition is greater than the minimum e required for cheats to replace cooperators (e*) ($2\left(d + a\left(-1 + \frac{1}{\beta}\right)\right) > \frac{2a-2a\beta+2c\beta}{\beta}$) provided $d > c$, which we have already assumed to be true (i.e., cooperators do better against other cooperators than against other cheats). Therefore, it is always possible for complementation to be large enough for cheats to replace cooperators, but small enough for the resulting population to have a lower mean fitness; a group-level advantage is not required for the evolution of multipartitism by cheating.

Finally, there is an alternative route to multipartitism in our model that does not involve cheating: the 2 cheats can invade and replace the cooperator even when the benefit to cheating

is relatively low ($b < d$) provided the complementation between cheats is sufficiently high. We do not focus on this route, because multipartitism would require very large levels of complementation benefit ($e > b$), and such large group benefits have been investigated more thoroughly elsewhere [1,4].

## Parameterisation details

To see if our model predicted the evolution of multipartitism via cheating under plausible parameter values, we next parameterised our condition for multipartitism to evolve via cheating ($e^* \geq \frac{2a-2a\beta+2c\beta}{\beta}$) using existing experimental data. Given empirical estimates of the relative fitness payoffs to cooperative viruses ($a$ and $c$), how common would coinfection have to be, and how large would the complementation between cheats ($e$) have to be, for multipartitism to evolve via cheating?

To estimate the degree to which cheats exploit cooperators within host cells ($c$), we used data from studies that compared the growth rates of cheats and cooperators. Some studies used single-cell approaches to explicitly quantify the ratio of cheat to cooperator genomes produced by coinfected cells; from these single-cell studies, we could take values for $c$ directly. Other studies compared the growth rates of cell cultures containing just cooperators, to cultures containing a mixture of cooperators and cheats. For these population-based studies, we estimated $c$ by comparing the output of productive virus from viral populations containing only cooperators, to viral populations that were started with equal frequencies of cooperators and cheats. Overall, we found that studies that used cheats containing large deletions, such as defective interfering genomes, resulted in $c$ values that were close to zero, indicating that cooperators receive negligible payouts when coinfecting alongside cheats. This is consistent with previous work that assumes such cheats are fully interfering [20,52,59–61].

We used an indirect method to estimate the fitness payout that cooperators receive when infecting cells on their own ($a$), relative to the fitness payout that cheats get when a cell is infected by the 2 types of cheat ($e$) (Fig 2). Estimating the ratio of $a$ to $e$ directly is not possible, because empirical studies to date have focussed on just 1 type of cheat, and so no estimates for $e$ are available. However, it is common for cells infected by multiple copies of the same viral genome to be more productive than cells infected by just a single copy, in which case $d > a$ [28–30,62,63]. Therefore, we estimated the ratio of $a$ to $d$, and then plotted the ratio of $e$:$d$ on the y-axis (Fig 3). Consequently, the y-axis of Fig 3 plots the degree of complementation between cheats ($e$) relative to the productivity of cells infected by 2 cooperators ($d$). To get an estimate of $a$:$d$ that is measured in the same way for all viruses, we use the ratio of non-infectious to infectious viral particles (the particle:PFU ratio). We compare the likelihood that at least 1 infectious viral particle reaches a cell infected by 2 viruses, compared to a cell infected by 1 virus ($a/d = \frac{1-x}{1-x^2}$ where $x$ is the particle:PFU ratio). When estimates varied for the particle:PFU ratio, we took the median value from the literature; when estimates were not available for a particular virus, we first looked for estimates for other viruses from the same family, and if those were not available, made the conservative assumption that the particle:PFU ratio was the highest possible [1]. This measure is potentially an overestimate of the $a$:$d$ ratio, since it ignores the potential for positive interactions between viral genomes beyond the chance of initially infecting a cell successfully; this would make our estimates more conservative with respect the evolution of multipartitism. In reality, it is also possible that these parameters are not independent. For example, if some benefit to viral coinfection exists, such that $d$ is large relative to $a$, then depending on the details of how this benefit arises, $e$ may be small relative to $d$. However, these relationships will depend on specific mechanistic details that are difficult to predict, so in Fig 3 we simply plot the relationships between parameter values and whether

multipartitism can evolve, without making further assumptions about how the parameters may relate to one another.

Parameter estimates are summarised in Table B in S1 Text and full details of the calculations are available with the supplementary material.

### Simulation lifecycle

In the simulation, we allow viral genomes to contain any number of genes (n_genes). We denote viruses encoding all n_genes as cooperators, and then, we allow for cheats to evolve that lack any number of the n_genes, in any combination. For example, if n_genes = 2, 1s represent genes present, and 0s represent genes absent, then cooperators are [1 1], and potential cheats are [1 0], [0 1], and [0 0]. We denote cheats encoding at least 1 gene as "partial cheats"; these are potentially able to complement one another and result in a multipartite virus population. In contrast, cheats encoding no genes whatsoever are "full cheats"; a population consisting entirely of full cheats would go extinct, but the possibility for full cheats could make it more difficult for partial cheats to invade [27].

We capture coinfection by assuming that host cells are infected by $k$ viral genomes each generation, where $k$ is drawn from a Poisson distribution with mean *lambda*, and the likelihood of each viral strategy being drawn depends on its relative abundance in the population as a whole. After host cells have been infected, we assume that the infection is successful only if the host cell contains at least 1 copy of every viral gene, and that the productivity of infected cells can increase with each additional set of viral genes (to capture $d > a$ in the analytical model). We then further modify this function to allow cells infected by partial cheats to be more productive, or less productive, than cells infected by cooperators, depending on a parameter $e$. Group benefits to multipartitism are therefore possible in the simulation when e > 1. To capture the differential competitive ability of cheats within cells, we then divide each cell's productivity among the viral genomes inside the cell in proportion to the number of genes encoded by each viral genome. This assumes that full cheats are the most competitive, followed by the different types of partial cheat, and cooperators least competitive; the magnitude of these competitive differences is controlled by a parameter $y$. The relative abundance of each viral strategy in the next generation is therefore determined by its share of the productivity of all the host cells that the strategy infected in the previous generation. Finally, we assume that the viral population is large and that mutation is unconstrained. Therefore, we capture mutation by assuming that mut_rate fraction of all viral strategies mutate each generation and that this fraction is then evenly distributed among every other possible viral strategy. When compared with previous models, this is a conservative assumption, making it more difficult for multipartitism to evolve, since we allow monopartite cooperators to re-evolve from cheats via mutation [12,21]. We determined that cooperators were extinct if their final frequency was less than 5 times the mutation rate.

To model the evolution of segmented genomes as well as multipartite ones, we then extended our simulation to allow multiple viral genomes to be incorporated inside the same virion. For reasons of tractability, we limited this extension to only consider the case where the viral genome contained 2 genes (n_genes = 2). We introduced a new parameter, "prop_single," which determines the fraction of virions from each infected cell that contain 1 viral genome (the remaining "1-prop_single" fraction of virions contained 2 viral genomes). We assume that virions are a fixed size, such that each virion can contain 2 copies of any of the viral cheat strategies, but can only contain 1 copy of the full-length cooperative genome. In this extension, we capture coinfection by assuming that host cells are infected by $k$ virions. We calculate the total productivity of infected cells and the relative competitive ability of genome strategies as

before. Within each infected cell, we then calculate the proportion of each type of virion produced by combining prop_single with the relative abundances of each type of viral genome strategy. For virions being filled by 2 viral genomes, we assume that each viral genome strategy is equally likely to copackage with any other viral strategy of the same length (i.e., packaging is unselective).

## Comparative dataset

To determine which viruses are known to produce defective interfering genomes, we: (1) searched Web of Knowledge and Google Scholar for the keywords "defective particle," "defective interfering particle," "autointerference," "Von Magnus," "therapeutic interfering particle," "TIP," and "DIP"; (2) searched the references of key reviews on defective interfering genomes [24,59,60,64–80]; (3) conducted forward- and backward-citation searches from the resulting articles. To ensure that we were only including viral cheats, we kept only records identified as defective interfering genomes, defined as truncated viral genomes generated from the wild-type, that were not able to infect cells on their own, and that interfered with the accumulation of the wild-type virus [20,24]. Overall, this resulted in a database of 49 viral genera known to produce defective interfering genomes.

To determine which viral genera are known to be multipartite, we took a list of known multipartite viral families from a recent review, and then, searched the database ViralZone for each genus within those families [4,81]. For each genus, we recorded the largest number of genome parts identified and classified it as multipartite if this number was $> 1$. When no information on the genus was available in ViralZone, we excluded that genus from further analysis. When ViralZone was inconclusive, we searched the primary literature for that genus to determine if it was known to be multipartite, recording the citation in our database. This resulted in 36 genera confirmed to contain multipartite species.

To classify viruses, we downloaded the most recent classification of viruses from the International Committee on Taxonomy of Viruses (ICTV) [82]. To attain a quantitative estimate for search effort, we iterated through each viral genus in the ICTV master list, and searched Scopus for primary research articles containing that genus name. After excluding viral genera with no research articles, and viral genera which are not assigned a realm, we were left with 644 viral genera.

## Code availability

We analysed the analytical model in Wolfram Mathematica 12, wrote the simulation in Matlab R2021a, and used R v4.1.0 to analyse the resulting data and produce the figures [83]. Figures produced in R made use of the R packages tidyverse, ggplot2, dplyr, plyr, and ColorBrewer [84–88]. Comparative data was collated in R v4.1.0. The authors would like to acknowledge the use of the Yale Center for Research Computing and the University of Oxford Advanced Research Computing (ARC) facility in carrying out this work: http://dx.doi.org/10.5281/zenodo.22558. Simulation results, plus all code used for mathematical analysis, simulation, cluster submission, comparative data collation, and figure production, are available at the following Open Science Foundation repository: https://doi.org/10.17605/OSF.IO/PBE4N.

## Supporting information

**S1 Text.** Supporting information file contains Figs A–E, and Tables A and B, together with a legend for each supplementary Figure and Table.
(DOCX)

## Author Contributions

**Conceptualization:** Asher Leeks, Geoff Wild, Stuart Andrew West.

**Data curation:** Asher Leeks, Penny Grace Young.

**Formal analysis:** Asher Leeks, Geoff Wild, Stuart Andrew West.

**Funding acquisition:** Asher Leeks, Paul Eugene Turner, Geoff Wild, Stuart Andrew West.

**Investigation:** Asher Leeks.

**Methodology:** Asher Leeks, Penny Grace Young, Paul Eugene Turner, Geoff Wild, Stuart Andrew West.

**Software:** Asher Leeks.

**Supervision:** Asher Leeks, Paul Eugene Turner, Stuart Andrew West.

**Visualization:** Asher Leeks, Paul Eugene Turner, Geoff Wild, Stuart Andrew West.

**Writing – original draft:** Asher Leeks.

**Writing – review & editing:** Asher Leeks, Paul Eugene Turner, Geoff Wild, Stuart Andrew West.

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
