## [Editor Report · Decision Letter 0]

6 Oct 2022

Dear Dr Leeks, 

Thank you for submitting your manuscript entitled "Cheating Leads to the Evolution of Multipartite Viruses" for consideration as a Research Article by PLOS Biology.

Your manuscript has now been evaluated by the PLOS Biology editorial staff and I'm writing to let you know that we would like to send your submission out for external peer review.

Once your full submission is complete, your paper will undergo a series of checks in preparation for peer review. After your manuscript has passed the checks it will be sent out for review. To provide the metadata for your submission, please Login to Editorial Manager (https://www.editorialmanager.com/pbiology) within two working days, i.e. by Oct 10 2022 11:59PM.

Kind regards,

Roli Roberts

Roland Roberts, PhD

Senior Editor

PLOS Biology

rroberts@plos.org

---

## [Decision Letter · Decision Letter 1]

19 Nov 2022

Dear Dr Leeks,

Thank you for your patience while your manuscript "Cheating Leads to the Evolution of Multipartite Viruses" was peer-reviewed at PLOS Biology. It has now been evaluated by the PLOS Biology editors, an Academic Editor with relevant expertise, and by three independent reviewers. 

In light of the reviews, which you will find at the end of this email, we would like to invite you to revise the work to thoroughly address the reviewers' reports.

As you will see below, the reviewers are mostly positive about your study, but each raises a few concerns that must be addressed before further consideration. As the Academic Editor summarised, "The paper is well written and interesting, yet I agree with the reviewers that especially the relation to the data should be made more explicit to warrant publication in PlosBio. I also think that Reviewer 1 is right that some generalizations of the model may not be too difficult. While I agree with Reviewer 2 that the manuscript should not be published in PlosBio in its *current* state, I am optimistic that the authors can improve it much by addressing the reviewer comments."

Given the extent of revision needed, we cannot make a decision about publication until we have seen the revised manuscript and your response to the reviewers' comments. Your revised manuscript is likely to be sent for further evaluation by all or a subset of the reviewers.

**IMPORTANT - SUBMITTING YOUR REVISION**

*Re-submission Checklist*

*Published Peer Review*

*PLOS Data Policy*

*Blot and Gel Data Policy*

Sincerely,

Roli Roberts

Roland Roberts, PhD

Senior Editor

PLOS Biology

rroberts@plos.org

REVIEWERS' COMMENTS:

Reviewer's Responses to Questions

PLOS authors have the option to publish the peer review history of their article (what does this mean?). If published, this will include your full peer review and any attached files.

Reviewer #1: Yes: Eugnee V Koonin

Reviewer #2: No

Reviewer #3: No

Reviewer #1:

[identifies himself as Eugene V Koonin]

The paper by Leeks et al explores a long-standing evolutionary problem, the origin of multipartite viruses (with implications for other genome segmentation phenomena). The authors develop a game theoretical model under which multipartitism can evolve through cheating alone, without any group benefits.

This approach is commendable because as a general principle, the possibility of non-adaptive origin of any phenomenon should be explored before adaptation is invoked. On the whole, the results of the model analysis are convincing. This is valuable and interesting work.

However, I believe the model fails to incorporate a fairly obvious and general advantage of genome segmentation. Different segments of multipartite genomes can be translated, transcribed and replicated simultaneously, resulting in increased rates of each of these processes, under the realistic assumption that the amounts of the enzymatic machineries involved are sufficient. I think a parameter capturing this enhancement of genome expression and replication should be included in any model. This will not invalidate the conclusion that cheating is at the root of multipartitism but is likely to help explain why genome segmentation is often fixed in evolution despite the costs. Many monopartite riboviruses of eukaryotes have evolved mechanisms of subgenomic RNA synthesis, circumventing the problem of expressing multiple genes from the same genome (mostly pertinent for viruses of eukaryotes in which indeed multipatitism is common). This is an evolutionary strategy that represent an alternative to multipartitism and probably should be discussed as such. Further, both multiparititims and sgRNA formation (and also programmed frameshifts in some riboviruses) provide the important opportunity for differentially regulating the production of virus proteins. It is certainly not chance that in the great majority of multipartite viruses, genes coding for replication system components and structural proteins are partitioned between different segments. All in all, I agree that cheating is at the root of multipartittims, but I believe that its widespread fixation in virus evolution also has adaptive components to it, I think this should be taken into account both in the model itself and its interpretation and discussion.

Another point pertains to viruses with multiple genome segments packaged within the same particle. In the current manuscript, these are just briefly mentioned towards the end of the Discussion. However, it seems to me that these can be smoothly included in the same modeling framework by replacing the cost and rate of coinfection with the corresponding values for genome segments assembly in virions. The choice of one strategy over the other during evolution will depend on the rate of coinfection, and this can be both modeled and gleaned from the available empirical data.

The rest of my comments pertain to presentation.

The current Introduction is rather verbose, but also somewhat too general. I believe the paper would benefit a lot from a more specific coverage of the actual data on virus multipartitism. Ideally, even a supplementary table containing such data across the virus realms, kingdom and phyla would strongly enhance the paper and help build the case for the major biological impact of multipartitism. 

In the Discussion, I believe it would be helpful to refer the Constructive Neutral Evolution concept that I believe is directly relevant for the non-adaptive origin of multipartititsm. 

The authors refer to the "lack of well resolved tree of viruses", citing Walker et al 2020. This is quite misleading as written. On the one hand, there is no such thing as an evolutionary tree of all viruses, but on the other hand, for many groups of viruses, the trees are quite well resolved This has to be rephrased.

The authors are unduly fond of the phrase "in principle" - I really think it should be minimized if not completely eliminated.

Reviewer #2:

In this article, the authors aim to better understand the existence conditions of multipartite viruses (hereafter:, mp viruses). They discuss existing hypotheses in the field, namely i) group benefits, ii) intragenomic conflicts between segments, and iii) small particles being favoured (e.g. bij persisting longer). The authors discuss how these hypotheses are limiting, as many mp viruses have 3 or more segments, so the opportunity costs are subsential while the imagined benefits are marginal. They support their skepticism well by stating that existing models requires incredible high rates of coinfection (up to 100) while in reality no more than 2-13 have been observed. The authors therefore challenge existing models by envisioning a hypothesis of their own: gene products of viruses are public goods, opening the door to cheats. Work has indeed shown that cheats can grow much faster than full genomes. As I understand it, mp viruses are, in the author's view, the result of "an evolutionary race to the bottom", or "tragedy of the commons". 

The manuscript is very well written, and I found only one minor error (see below). However, while I am happy to consider the hypothesis by the authors, I find their methods and the analysis of the models lacking. My biggest concern is that that authors phrase an inevitable consequence of cooperator-defector dynamics to be the one and only explanation needed to understand mp viruses. At numerous (at least 5) points in the text, they found it necessary to discuss how their model is superior because it does not require group benefits. In one occasion in particular, it was even phrased in such a way as to suggest no further work was needed: "Consequently, our work does not require new kinds of mechanism to be uncovered in order to explain multipartitism, nor does it require searching for elusive group benefits to multipartitism." In my opinion, this is in very poor form, where it could even suggest that "we know it all now". This is not productive, and not helpful to scientific discourse. 

The authors move on to make multiple predictions based on their game theoretical model, where mp evolves if:

1. coinfection is common 

2. advantage of cheating is high

3. high e (which honestly, is an assumption that requires unpacking!)

4. cooperators are strongly outcompeted within their cells (which may be a strong assumption, but I grant you this given the 10,000 fold benefit discussed in the intro)

5. cooperators do worse alone than with another cooperator (is this a fair assumption?)

Generally, I found the predictions marginally insightful, as most of them came down to a cost-benefit analysis without any mechanistic insight as to what drives the costs- and benefit-parameters in the first place. For example: "Our model predicts when cheats are highly competitive relative to cooperators [...], cheats evolve. This is not particularly insightful. The authors attempt to support these insights with figure 3, but to make this analysis somewhat convincing, it should also contain examples of clearly non-mp viruses. The way the data is presented here, any virus could be either mp or non-mp, as the parameter range shown is sufficiently large as to capture all possible outcomes. Perhaps the most insightful one was a brief sentence all the way at the end of the discussion, with a reference to figure S7. The figure itself however is not very well thought through (having a boolean variable on a continuous colour gradient), but I found this figure very interesting, if only it gained half the attention of the rest of the manuscript. I think your work could be much more interesting if you'd consider going into more detail with S7, and perhaps also investigate why many viruses are NOT mp, considering how your analysis suggests this to be a very prominent attractor of virus evolution. I would the analysis w.r.t. the "cheat load" is one that would truly bring your work to the next level (so, S7, but also what is discussed at L397 onwards)

While I actually support the angle that the author take, and found the writing incredibly well thought out, I would not recomment this work to be published in its current state. I want to clarify that I find it really difficult to write such a negative review, and that this is simply my best attempt at trying to help to improve the manuscript for future submissions. I wish the authors the best of luck! 

Minor points:

1. Line 96 repeats line 85, quite quickly after the first claim of fitness benefits of cheaters. I don't mind some repetition, but this was so shortly after eachother that I think it can be merged. 

2. Figure 1 has very detailed genomes inside the hexagons while they are really small. Perhaps consider moving the legend some place else so the hexagons can be a bit bigger. Or alternatively, have less detailed genome illustrations to make the figure more readable. 

3. Payout d is not discussed in the main text, although I could infer it from the figures. Now, the main text discusses payout a, b, c, and e (and not d), which should be resolved. 

4. The way the simulations work is not discussed well enough in the main text (or methods). Please provide a summary of appendix 3 in the main text. 

5. "No other factors need to be invoked." similar to my major concern: this suggests the work is "complete", which it of course isn't. There is plenty to still unpack here. 

6. S7 is called S6 in the caption. 

Reviewer #3:

This manuscript addresses the important and unresolved question of the evolution of multipartite viruses. That cheaters can be at the origin of the evolution of such viral systems is not new, but the key results provided here are indeed novel and original. Some of the conclusions appear highly interesting, contrasting with previous models and current state of the art: i) even highly multipartite viruses can evolve when multiplicity of infection is reasonably low, and ii) with no requirement for group-level benefit.

I have two related major general comments and a list of specific comments that should be addressed to improve the quality of the paper, including for readers that are not deep in theoretical approaches (just like me). 

Major comments:

1-In the discussion, the authors indicate that the selective advantage of cheats explains genome fragmentation and that the work presented in the manuscript is thus relevant not only for the evolution of multipartite viruses but also for segmented viruses encapsidating their genome segments together in a single viral particle. While true, this is a serious concern because most earlier attempts at explaining the evolution of multipartite viruses were in fact addressing the genome segmentation but not specifically the separate encapsidation of each segment, which is the essence of multipartitism. In this work, one may wonder whether what is addressed is indeed the individual encapsidation of the segments, as claimed in the title and all along the text through the term "multipartite", or less specifically the genome segmentation. The modelling presented here is in a world where one nucleic acid molecule (whether full length or cheat) is individually encapsidated. What would happen in a slightly more realistic world where encapsidation of more than one viral nucleic acid molecule would be possible, e.g., more than one cheat. With such an additional possibility, could multipartitism evolve, or would the so-called segmented virus packaging several segments together systematically take over?

2-Perhaps related to the above comment, the attempt at extending the results/conclusions presented here to very different systems such as integrated phages, plasmids, and bacterial endosymbionts, is troublesome. It is confusing because these different biological systems are presented as if they all face the same cost ("despite these costs multipartite genomes are found widely throughout nature"). They do not. What they have in common is the fragmentation of the genome but not the separate encapsidation and so not the separate spread of the distinct segments. Endosymbionts of cicadas have split lineages with distinct genome parts, but they are in the same insect cells or bacteriocytes. Each lineage exchange mandatorily with the host but not directly with another lineage, or at least this would need demonstration. Plasmids recombine their replication origins and or can occasionally highjack the conjugation system encoded by another plasmid, but they do not necessarily reciprocally provide complementary functions, so direct comparison with multipartite viruses is arguable, in the least. Integrated bacteriophages are scattered in the same host genome and vertically transmitted together.

The problem is that multipartitism is not clearly defined from the start (first lines of the introduction) as several genome segments (interdependent) encapsidated and propagated/spread separately from cell-to-cell and host-to-host; Or do the authors have another definition? I am not sure that the incurred cost is the same for multipartite viruses, segmented viruses, integrated phages, plasmids, and bacterial endosymbionts, I even clearly doubt it. 

These two points must be fixed to make sure that the submitted manuscript specifically addresses the puzzling feature of multipartite viruses, that is the individual encapsidation of each of their genome segments.

Specific comments

-Line 39: "…segments must independently reach the same host…" the reference Di Mattia et al PNAS 2022 also appears appropriate when mentioning host-to-host transmission of segments. More importantly, though it may be implicit in the text, it is important that all readers understand that multipartitism means that each viral genome segment is encapsidated individually/separately. The following sentence may be slightly modified for "This type of genome segmentation, with each segment packaged separately, is termed multipartitism and entails substantial costs; most infections…"

-Lines 55-57: I do not think that Gallet et al 2022 state that this benefit allowed the evolution of multipartitism. In multipartite viruses, what they report could be a benefit, this does not mean that it is what drove their evolution.

-Line 57-61: I do not understand how the example published by Ojosnegros and colleagues illustrates minimal conflict between different genome segments?

-Lines 66-67: is it 100 viral genomes or 100 viral particles? There will be other similar questions below and it is very important that the authors are clear in the terminology they use when speaking about a viral genome, a genome segment, a virus, a virus particle. This is mandatory to clearly understand the issues of co-infection or MOI in this paper.

-Line 109: "We examined the theoretical feasibility of the cheat hypothesis for the evolution of multipartitism and then tested it empirically": I do not see the empirical testing in this paper. Please explain.

-Line 133-135: what is called a viral genome here? The cooperator monopartite, the sum of D1+D2 or only D1 or D2. 

-Line 154 and payoff matrix: in case of infection by one of each cheat or by the cooperator and one cheat, same payout "e" and "b" for each cheat, but this is likely a rare situation. Would it change something to the model output if payout would be different for D1 and D2?

-Figure 3: It appears bizarre to parametrize the model with data from polio, VSV, Rabies, and bunyawera, which are monopartite animal viruses. Of note however, for rhabdoviruses and bunyavirales, some multipartite species do exist in plants. 

-Line 233: "we allowed >2 viruses to co-infect cells": again, what means "viruses" in this sentence? (IDEM line 258)

-Lines 263-269: yes! but this largely depends on the benefit of cheats themselves (b and e) relative to that cooperator. What are the values leading to multipartitism, are they reasonable?

-Same lines: I am wondering how or when "e" can be considered a group benefit? when e>c, a or d. But could it be when e>b? Perhaps this is indicated somewhere.

-Figure 4: why does crash occur only at high MOI? This could be explained either here or in the discussion.

Lines 299-301: I have a problem with the data on which this is based. What exactly is important? Is it the rate of production of defectives or their capacity to accumulate in the system? These may be totally distinct, and the data collected here could be biased in that sense. Let's suppose that all viruses rampantly produce tones of defectives but only in some cases one defective "species" becomes sufficiently frequent to be detected. The collectively huge amounts of myriads of distinct defectives would be mostly overlooked (unless long read sequencing is analyzed). To date, only scattered studies are available, and most do look at defectives that accumulate, not at their rate of production. But perhaps the two are linked. I don't know. I acknowledge the fact that the authors are already cautious about this conclusion of Figure 5, but more discussion is required. When going more locally in the viral clades, some closely related viral taxa have both mono and multipartite member species and so the correlation shown in Figure 5 appears to depend on the chosen scale.

-Line 331: what means "viral genome" here?

-Lines 332-336: It seems that multipartite viruses with 2 or 3 segments can evolve at MOI that are even compatible with values reported in animal viruses, or phages, and yet they do not exist there. Please comment on this in the discussion, specifically for multipartite viruses with a low number of segments.

-Line 354: I do not understand what means "identify conditions under which multiple different types of cheat are possible"

-Lines 363-364: this is rather a matter of number of generations, the time is not key, 

The authors seem to consider this could arise in a few generations but then the evolved multipartite would meet the monopartite ancestor all the time in the outworld.

-Lines 365-366: Could the authors be more specific on the estimate of transmission bottleneck comparison between plants and animals because they may not be so different, there are not a lot in the literature and so actual numbers could be discussed.

-Lines 384-385: Do the authors imply that reversion to monopartite is possible? or that monopartite related virus species would outcompete those that have evolved multipartitism. This point should be explicit.

-Lines 415-421: Yes, segmented viruses are concerned and encapsidating together several segments would facilitate co-infection. Both specific and non-specific encapsidation of segments exist in segmented viruses, and this may have a distinct impact on "co-infection". It would be interesting to see how a multipartite virus would do when competing with similarly segmented genomes but packaged together. It is even a key question for the conclusion of this manuscript.

-Lines 415-418: What would lead evolution into multipartite forms rather than segmented forms? Please comment.

-Lines 423-430: This goes more and more towards segmentation (unfortunately) rather than multiple encapsidation and separate propagation of segments or genome fragments. Please see the related Major comments.

-Lines 439-440: A more optimistic view would present those conflicts has creating opportunities for the transient appearance of more complex genetic systems that may then unveil emerging beneficial properties. Such beneficial properties (which do not exist for monopartite ancestors) could in some cases allow the maintenance of multipartitism. Multipartitism would thus evolve from conflicts but be maintained by emerging properties in the system.

-I do not understand Figure S7 (that is mistakenly labelled Figure S6 I think). Either the legend should be re-written, or the Figure organized differently. What are the "0" and "1" down the graph? What do the 1 to 10 numbers on the top indicate?

---

## [Decision Letter · Decision Letter 2]

6 Mar 2023

Dear Dr Leeks,

Thank you for your patience while we considered your revised manuscript "Cheating Leads to the Evolution of Multipartite Viruses" for publication as a Research Article at PLOS Biology. This revised version of your manuscript has been evaluated by the PLOS Biology editors, the Academic Editor and the original reviewers.

Based on the reviews, we are likely to accept this manuscript for publication, provided you satisfactorily address the remaining points raised by the reviewers. Please also make sure to address the following data and other policy-related requests.

IMPORTANT. Please attend to the following:

a) Please address the remaining requests from reviewer #3.

b) Please provide a blurb, according to the instructions in the submission form.

c) Please cite the location of the data and code clearly in all relevant main and supplementary Figure legends, e.g. “This Figure can be generated using the data and code in https://osf.io/y623v/XXXXX"

We expect to receive your revised manuscript within two weeks. 

*Published Peer Review History*

*Press*

Sincerely,

Roli Roberts

Roland Roberts, PhD

Senior Editor,

rroberts@plos.org,

PLOS Biology

DATA NOT SHOWN?

REVIEWERS' COMMENTS:

Reviewer #1:

[identifies himself as Eugene V Koonin]

I found the revisions made by the authors to be satisfactory and a substantial improvement, making the paper highly convincing.

In particular, by including an interesting discussion of potential group benefits of multipartitism, I believe the authors address the comments of reviewer 2 regarding biases in the original presentation.

I have no further suggestions.

Reviewer #2:

[identifies himself as Bram van Dijk]

Dear authors,

Your revisions are really thorough, to the point that I really enjoyed reading your responses to even my most hard criticism. I also am very happy with the new figures that were added to get a little bit deeper into the topic.

I have no further comments, and I am in favour of publication. 

Best,

Bram

Reviewer #3:

Leeks et al 2022 R2

I do consider that the authors have mostly satisfactorily responded to my comments. Therefore I have no additional comments, but some very minor points

-Line 37 (but also elsewhere including the Abstract): I am not sure what "independently" or "separately"' means here and in the abstract. For nanoviruses, that segments can move separately within plant and also between plants have some experimental support. In rift valley fever bunyavirus, that distinct particles containing incomplete sets of segments can-also move separately cell-to-cell and even into insect vector, co-infect and reconstitute a full infection, also received experimental support recently. However, it does not imply that they HAVE TO move separately or independently to successfully infect. This sentence can be slightly modified just to nuance this point.

-Line 45-50: I am still troubled by the fact that the authors want to generalize their findings or investigation to plasmids, integrated phages and endosymbionts. Their model applies to viruses and I am not sure I can clearly see how and whether it could similarly apply to these other organisms, whether they are really "analogous" type of genome fragmentation. I understand that the authors really wish to discuss this point and so I suggest to remove it from the Introduction and rather keep the part in the discussion. Also, the authors may want to delete the terms -"analogous" types of genome fragmentation- or -"similarly" fragmented-. "Other types of genome fragmentation" and "are also fragmented" may be more appropriate

-Line 52: focus or rely ?

-Line 326 Section on segmented viruses: It remains unclear to me if multipartitism can evolve and get fixed when segmented viruses (encapsidating segments together) are allowed. Could multipartite outcompete segmented in some instances ?

-I have the feeling that many predictions from Table 1 are not supported by available data but, I agree with the authors that most have not been formally tested and these predictions have the merit to now exist and, in that sense, I agree they are interesting.

-Lines 487-488: Transmission by insect is most often (if not always) shown to induce strong bottlenecks. The sentence in these line apparently states the contrary? "….resulting in relatively wider bottleneck" relative to what ? The examples given appears to be chosen to make the authors point, not reflecting the state of the art. There are vector transmitted animal viruses with large bottlenecks at transmission (mosquito transmitted for example). The bottlenecks during between host transmission of IAV and other HIV / HCV are not always small. In addition, the value given for Cauliflower mosaic virus, is not a value for between host transmission but the value for intra-plant cellular MOI ? Values for vector transmission bottlenecks for potyviruses and cucumoviruses are smaller I think. 

Perhaps, one way out of this could just be to argue that these are values from single aphid vector and that the bottlenecks may be enlarge by several aphids visiting the plants in the field (this type of information is unfortunately poorly documented but the possibility of several aphids visiting the same plant has been often mentioned)

---

## [Editor Report · Decision Letter 3]

22 Mar 2023

Dear Dr Leeks,

Thank you for the submission of your revised Research Article "Cheating Leads to the Evolution of Multipartite Viruses" for publication in PLOS Biology. On behalf of my colleagues and the Academic Editor, Claudia Bank, I'm pleased to say that we can in principle accept your manuscript for publication, provided you address any remaining formatting and reporting issues. These will be detailed in an email you should receive within 2-3 business days from our colleagues in the journal operations team; no action is required from you until then. Please note that we will not be able to formally accept your manuscript and schedule it for publication until you have completed any requested changes.

Sincerely, 

Roli Roberts

Senior Editor

PLOS Biology

rroberts@plos.org